# Impact of Anticoagulants in Reducing Mortality and Disability in Cardioembolic Stroke Patients

**DOI:** 10.3390/medicina58101323

**Published:** 2022-09-21

**Authors:** Kristaps Jurjāns, Marija Cērpa, Alise Baborikina, Oskars Kalējs, Evija Miglāne

**Affiliations:** 1Department of Neurology and Neurosurgery, Riga Stradins University, 16 Dzirciema Street, LV-1007 Riga, Latvia; 2Department of Doctoral Studies, Riga Stradins University, 16 Dzirciema Street, LV-1007 Riga, Latvia; 3Department of Neurology, Pauls Stradins Clinical University Hospital, 13 Pilsonu Street, LV-1002 Riga, Latvia; 4Faculty of Continuing Education, Riga Stradins University, 16 Dzirciema Street, LV-1007 Riga, Latvia; 5Department of Internal Medicine, Riga Stradins University, 16 Dzirciema Street, LV-1007 Riga, Latvia; 6Latvian Centre of Cardiology, Pauls Stradins Clinical University Hospital, 13 Pilsonu Street, LV-1002 Riga, Latvia

**Keywords:** atherothrombotic stroke, cardioembolic stroke, stroke functional outcome, stroke mortality, anticoagulants, antiplatelet agents

## Abstract

*Background and Objectives:* Stroke is currently the second most common cause of death and disability-adjusted life years worldwide. Previous studies have determined that cardioembolic stroke is associated with higher mortality. Our aim is to compare the long-term outcome and mortality of atherothrombotic, cardioembolic stroke patients and patients taking direct oral anticoagulants (DOACs), and to demonstrate that adequate treatment with DOACs is associated with better results. *Materials and Methods:* In our retrospective study, we collected the data of ischemic stroke patients who were treated at P. Stradins Clinical University Hospital, Riga, Latvia, Stroke Unit, in the year 2017. In the present study, we analyzed this information to assess the patients’ demographic and clinical data, vascular risk factors, functional and neurological evaluation results, and the use of anticoagulant therapy. Stroke survivors were followed-up via telephone at 30/90/180/365 days and 4 years after being discharged from the hospital. The Latvian version of the National Institutes of Health Stroke Scale (NIHSS-LV) was used to evaluate patients’ neurological outcomes at discharge, and patients’ functional outcomes were evaluated using the modified Rankin scale (mRS). The collected data of the patients were separated into three groups according to the stroke subtype and use of direct oral anticoagulants. *Results:* A total of 654 ischemic stroke patients were admitted to the hospital in the year 2017. Of all the strokes included in the study, 262 presented an atherothrombotic etiology and 392 presented a cardioembolic etiology. The median age of the patients in the study was 76 years (IQR: 67–83). The median age of patients in the atherothrombotic stroke group was 71 years (IQR = 64–79), in the cardioembolic stroke group it was 79 (IQR = 72–84), and in the DOAC group it was 75 years (IQR = 69–82), respectively. At the period of four years, of all the atherothrombotic stroke survivors 14 (10.5%) had a severe disability, and 64 (48.1%) did not survive. However, 12 (4.1%) of the cardioembolic stroke survivors were severely disabled and 37 (12.5%) had died. In the group of patients taking DOACs 6 (4.5%) had a severe disability and 17 (12.9%) did not survive. In all the patient groups, the leading cause of death was due to severe disability (22%), followed by recurrent cardioembolic events (8%). *Conclusions:* Previous studies until now have concluded that cardioembolic stroke is associated with higher mortality and an unfavorable functional outcome. In our study, the cardioembolic stroke group and the DOAC group had a statistically significant higher percentage of patients with congestive heart failure and older age, but their long-term mortality was lower and they achieved independence more often than the atherothrombotic stroke patients. The proper use of anticoagulants shows great improvement in long-term survival rate and functional outcome.

## 1. Introduction

Stroke is currently the second most common cause of death and disability-adjusted life years worldwide. The likely explanation for this is the increase in and aging of the world’s population in addition to decreased death rates globally in recent decades [1].

Previous studies have determined that cardioembolic stroke is associated with higher mortality rates as it is, in general, more severe and prone to early and long-term recurrences. In a study by Ferro (2003), in-hospital mortality in patients with early embolic recurrences (within the first seven days) was 77% [2,3]. This is associated with higher volumes of more severe baseline hypoperfusion, leading to greater infarct growth rates and a higher risk of hemorrhagic transformations (up to 71% of cardioembolic strokes) and secondary hematomas, which results in various effects and clinical deterioration [4,5,6]. There is also an increased risk in patients with atrial fibrillation for a large vessel occlusion with a following more severe neurological impairment; hence a significantly poorer self-perceived Health-Related Quality of Life (HRQoL) at 90 days as presented in a study by Masiliūnas (2022) [7]. The recurrence of a cardioembolic stroke can be prevented by regularly taking oral anticoagulants [8].

The data on long-term mortality rates for cardioembolic strokes are limited. In a recently conducted Latvian study, it was concluded that alcohol abuse as a pre-stroke risk factor, poststroke urinary incontinence as a neurological symptom and the dependence on grooming as a factor of disability were associated with earlier mortality in the first seven years after the occurrence of a stroke [9]. The present study aims to compare the functional outcome and mortality in patients with atherothrombotic and cardioembolic stroke and patients taking direct oral anticoagulants (DOACs). Previous studies have concluded that cardioembolic stroke is associated with higher mortality and poorer functional outcome; however, we aim to show that proper therapy with anticoagulants in patients with cardioembolic stroke and non-valvular atrial fibrillation (VNAF) yields better results—lower mortality and better functional outcome.

## 2. Materials and Methods

In this retrospective study, we included the data of ischemic stroke patients who were admitted to Pauls Stradins Clinical University Hospital, Riga, Latvia, in 2017. We collected the patients’ clinical data using the local stroke registry, including patients’ demographic data, vascular risk factors, and functional neurological evaluation results. The Latvian version of the National Institutes of Health Stroke Scale (NIHSS-LV) was used to evaluate patients’ neurological outcomes at discharge, and the modified Rankin scale (mRS) was used for assessing the functional outcome [10,11,12]. The trial of ORG 10172 in acute stroke treatment (TOAST) classification was used for the determination of stroke etiology [13]. Only atherothrombotic and cardioembolic stroke patients with VNAF were included in our study. We excluded patients with other ischemic stroke subtypes according to TOAST criteria (small-vessel occlusion, stroke of other determined etiology, and stroke of undetermined cause), as well as hemorrhagic stroke patients. Patients with other possible cardioembolic causes (mechanical prosthetic valve, mitral stenosis with atrial fibrillation, left atrial/atrial appendage thrombus, sick sinus syndrome, recent myocardial infarction (<4 weeks), left ventricular thrombus, dilated cardiomyopathy, akinetic left ventricular segment, atrial myxoma, and infective endocarditis) were also excluded. Patients’ contact information and patients’ or their relatives’ permission to participate in the study were necessary to perform the follow-up.

Stroke survivors were followed up by telephone interviews at 30, 90, 180, 365 days, and 4 years after discharge. Patients’ functional outcome was evaluated using the adapted version of The Rankin Focused Assessment-Ambulation (The RFA-A) [11]. We asked standardized questions about patients’ usage of prescribed medication and recurrent cardioembolic events. If the patient had died, the cause of death was identified by the patient’s relatives. Patients were divided into 3 groups: patients with atherothrombotic stroke, patients with cardioembolic stroke, and separately patients with cardioembolic stroke taking DOACs. mRS range from 0 to 2 was identified as a favorable outcome, 3 as a successful outcome, and from 4 to 5 as an unsuccessful outcome, without mentioning those who did not survive.

Microsoft^®^ Office Excel 2016 (Redmond, WA, USA and IBM Statistics 24 (Armonk, NY, USA) were used to analyze the collected data. Descriptive statistics were applied for continuous variables (median, quartiles) and categorical variables (numbers and percentages). The Chi-square test was used to compare categorical variables, Mann–Whitney test was used for continuous variables. The Kaplan–Meier method and Cox proportional hazards were used to compare 4-year survival rates among the groups. 95% confidence intervals (CI) were calculated for all 3 patient groups.

The study’s ethical aspects were approved by The Ethics Committee of Riga Stradins University (ethical committee approval, nr. 23/29 March 2018). Verbal permission of participation was requested while contacting patients or their relatives during a telephone interview.

## 3. Results

### 3.1. Patient Characteristics

In the course of the study, a total of 654 ischemic stroke patients were admitted to Pauls Stradins Clinical University Hospital, Department of Neurology, Riga, Latvia. Out of 262 (40.1%) patients with atherothrombotic stroke 130 (49.6%) were females and 132 (50.4%) males. A total of 392 (59.9%) cases were identified as cardioembolic strokes. Out of all the cardioembolic stroke sufferers, 182 (85.4%) patients were receiving DOACs, of which 122 (67%) were females.

The median age of the patients in the study was 76 years (IQR = 67–83). The median age of patients in the atherothrombotic stroke group was 71 years (IQR = 64–79), in the cardioembolic stroke group was 79 (IQR = 72–84), DOAC group was 75 years (IQR = 69–82) respectively. Both gender and average differences were statistically significant between the atherothrombotic stroke group and cardioembolic stroke group, and the atherothrombotic group and DOAC group (*p* < 0.01; *p* < 0.001).

The most common ischemic stroke risk factors among the groups were arterial hypertension: 552 (84%), dyslipidemia: 308 (47%), congestive heart failure: 248 (38%), recurrent stroke or transitory ischemic attack (TIA): 179 (27%), diabetes mellitus: 116 (18%), coronary heart disease: 105 (16%). We found that 65 (10%) of all patients had a previous heart attack, 29 (4%) patients had chronic kidney disease and 24 (4%) were identified as smokers. There was a statistically significant higher amount of recurrent stroke or TIA and dyslipidemia in the atherothrombotic stroke group, as well as more patients were smokers in this group.

Of all the nonvalvular atrial fibrillation patients, 277 (70.1%) were identified with a permanent atrial fibrillation form and 115 (29.9%) with paroxysmal form.

In the atherothrombotic stroke group, 59 patients received the revascularization therapy: 44 (16.8%) patients were treated with intravenous thrombolysis (IVT) using tissue plasminogen activator (t-PA) Alteplase, 3 (1.1%) underwent mechanical thrombectomy (EVT), and 12 (4.6%) received IVT and EVT. In its turn, 112 patients from the cardioembolic stroke group underwent reperfusion therapy: 69 (17.6%) received IVT, 13 patients (3.3%) were treated with EVT, and 30 (7.7%) with IVT + EVT. In the DOAC group, a total of 71 patients received IVT and/or EVT: 46 patients (25.3%) were treated with IVT, 8 (4.4%) with EVT and 17 (9.3%) with a combination of both methods. No significant statistical difference in the choice of treatment was observed between the atherothrombotic and the cardioembolic stroke groups (*p* = 0.103); however, there was a statistically significant difference comparing the atherothrombotic stroke group and the DOAC group (*p* = 0.001). See Table 1.

Internal carotid artery (ICA) stenting during acute EVT or IVT + EVT was performed in 9 (3.4%) atherothrombotic stroke group patients, 2 (0.5%) patients in the cardioembolic stroke group, 1 (0.5%) patient in the DOAC group. The difference in ICA stenting between the atherothrombotic and cardioembolic stroke groups was statistically significant (*p* = 0.009), as well as the difference between the atherothrombotic stroke group and the DOAC group (*p* = 0.05).

Out of all patients from the atherothrombotic stroke group, 111 (42.4%) patients had hemodynamically severe ICA stenosis, and 19 (7.3%) underwent an ICA endarterectomy. At the time of discharge, 160 patients were taking antiplatelet monotherapy, while 102 (43.0%) patients took dual antiplatelet agent therapy (DAPT) for at least 21 days as secondary ischemic stroke prophylaxis. Although it is estimated, that patients receiving DAPT have a higher risk of bleeding complications, we did not observe a higher risk of DAPT-associated bleeding in this group of patients.

The median NIHSS-LV on admission was 6 (IQR = 4–10) in the atherothrombotic stroke group, 8 (IQR = 4–15) in the cardioembolic stroke group, and 7 (IQR = 3–13) in the DOAC group. The median NIHSS-LV at discharge was 3 (IQR = 2–6) in the atherothrombotic stroke group, 3 (IQR = 2–9) in the cardioembolic stroke group, and 3 (IQR = 1–5) in the DOAC group, respectively. The median NIHSS-LV at discharge was higher in the cardioembolic stroke group, and the best improvement result was observed in the DOAC group. This difference among the groups was statistically significant (*p* < 0.05).

The patients’ demographic characteristics in all 3 groups are shown in Table 2.

### 3.2. Survival Analysis

The crude probability of survival in 654 ischemic stroke patients from all groups is presented in Figure 1 and Figure 2. The combined mortality after the discharge from the hospital was 10.3%, 7.2% in 30 days, 6.8% in 90 days, 4.5% in 180 days, 4.6% in one year, and 23.5% after 4 years. The median survival was 1411 days (IRQ = 1748–109) in the atherothrombotic stroke group and 316 (IQR = 1692–9) in the cardioembolic stroke group. Comparing the atherothrombotic stroke group with the DOAC group, median survival was 1341 (IQR = 1744–100) in the atherothrombotic stroke group and 1754 (IQR = 1787–522) in the DOACs group, respectively. The median survival time was significantly longer in the atherothrombotic stroke group comparing to the cardioembolic stroke group (log-rank test, χ^2^ (1) = 16.751, *p* < 0.001), but shorter than in the DOACs group (log-rank test, χ^2^ (1) = 3.486, *p* = 0.05).

The cardioembolic stroke patients (including DOACs group) were statistically significantly older than the atherothrombotic stroke patients, and the use of antithrombotic treatment was statistically significant between patient groups. The Cox proportional hazard regression model was performed to compare the patients’ groups, including age and reperfusion therapy as a probable cofactor. Results are shown in Table 3.

Comparing the atherothrombotic stroke patient group and the cardioembolic stroke patient group using age as a possible cofactor, the hazard ratio (HR) was 1.059 according to the 95% confidence interval (CI), which was statistically significant (*p* < 0.001). The HR between the atherothrombotic stroke group and the DOAC group was 1.045 (*p* < 0.001). When using reperfusion therapy as a second outcome impacting cofactor, the HR comparing atherothrombotic and cardioembolic patient groups was 0.844 (*p* = 0.1), while comparing atherothrombotic and DOACs group, HR was 0.540 (*p* < 0.001). It is not possible to determine if a specific type of antithrombotic therapy was more effective, or if all of the methods played an important role in the outcome of the stroke survivors because of the insufficient number of patients in the EVT and IVT + EVT groups.

After the adjustment for confounders, the secondary prevention showed significant statistical importance in cardioembolic stroke patients and patients taking DOACs, and with statistically significantly fewer hazards than patients with atherothrombotic stroke.

### 3.3. Functional Outcome Analysis

The functional outcomes obtained during the timespan of the study in the patients’ groups are presented in Figure 3, Figure 4 and Figure 5.

Upon their discharge, 37% atherothrombotic stroke patients had favorable functional outcome (mRS 0–2), 17.9% had successful functional outcome (mRS 3), 35.5% had unsuccessful functional outcome (mRS 4–5) and 9.5% patients died (mRS–6). The number of favorable functional outcome cases continued growing until the first year after discharge. Then it decreased from 67.8% to 35.3%, while the mortality rate rapidly increased from 11.2% to 48.1%.

Out of 392 cardioembolic stroke patients, at the discharge day, almost half of the patients reached a favorable functional outcome. Intrahospital mortality was slightly higher in this group than in the atherothrombotic stroke group, being equal to 11% compared to 9.5%. The number of favorable outcome cases continued to grow during the 4 year period. At the same time, mortality rates decreased during regular follow-ups, but increased in long-term follow-ups, being equal to 12.5%.

In patients receiving DOACs, more than half of patients reached mRS 0–2, and only 2.3% died while staying in the hospital. Similar functional outcome changes as in the cardioembolic stroke group were observed in these patients; however, 4 year mortality was higher—12.9% compared to 12.5%.

In all the patients’ groups, the leading cause of death was due to severe disability (22%), followed by recurrent cardioembolic events (8%). A total of 1% of all patients died due to COVID-19 infection.

## 4. Discussion

Our study analyzed long-term functional outcomes and mortality of atherothrombotic and cardioembolic ischemic stroke patients, as well as the most common risk factors. We additionally selected the patients taking DOACs from all cardioembolic ischemic stroke survivors to compare if they had better functional outcomes and survival than other patients with NVAF. Cardioembolic strokes made up 59.9% of all ischemic strokes while the atherothrombotic subtype comprised 40.1%. This data is similar to the latest studies, showing the increase in cardioembolic strokes worldwide [14,15]. A similar increase in the incidence of cardioembolic ischemic stroke was reported in another Latvian Stroke Unit [16]. Our study consisted of the systematically collected data, as well as multiple follow-ups conducted via telephone, of 654 patients who were treated at our hospital’s Stroke Unit in the year 2017. The ischemic stroke incidence in Latvia is 131–150 per 100,000 people [1] and our center is only one of the seven available stroke units in Latvia [8]; therefore, the data of the patients does not represent the entire stroke population in our country [14].

Guglielmi et al. compared atherothrombotic and cardioembolic functional outcomes in ischemic stroke patients and reported better functional outcomes in the atherothrombotic patient group—mRS 3 (IQR = 1–5) versus 4 (IQR = 2–6) in the cardioembolic patient group. Satisfactory functional outcome after 90 days (defined as mRS 0–2) reached 46% in the atherothrombotic ischemic stroke patient group and only in 35% of all cardioembolic ischemic stroke survivors [17]. Jurjans et al. stated that in patients receiving antiplatelet agents, only 3.0% of patients reached satisfactory functional outcomes (mRS 0–2) in 5 years, compared to 30.5% in patients receiving direct oral anticoagulants [8].

Our results show that of all atherothrombotic ischemic stroke patients, 35.3% reached a satisfactory functional outcome (mRS 0–2) compared to 79.1% in the cardioembolic ischemic stroke patient group and 77.3% in the DOAC group. Сardioembolic ischemic stroke patients and DOAC patients are more likely to get a satisfactory functional outcome rather than patients with large vessel occlusion. Identical results were reported by Guglielmi et al. [17]; however, there are studies with opposite statements [18].

In our study, 48.1% of the atherothrombotic group had died in 4 years compared to 12.5% in the cardioembolic patient group. In the DOAC group, only 12.9% died within 4 years. This might be because a meta-analysis of DOACs determined non-inferiority for the prevention of ischemic stroke and superiority for the rates of hemorrhagic stroke [15,16]. In the atherothrombotic stroke group, a satisfactory functional outcome was obtained for only 35.3% of patients. The possible influential factors might be ‘breakthrough’ cryptogenic strokes and aspirin or clopidogrel resistance, which could result in recurrent strokes [19,20].

In a previous study by Jurjans K. et al. (2019), it is concluded that anticoagulant use in secondary prevention predicts a better functional outcome and higher survival rate in patients with severe cardioembolic stroke due to NVAF [8].

Halkes et al. found that the most common mortality factors were recurrent cardioembolic events (myocardial infarction, ischemic stroke, pulmonary artery thromboembolism, etc.) following complications of immobility (deep vein thrombosis, pressure wounds, pneumonia, etc.) [21]. Due to anticoagulant use, the cardioembolic stroke patient group was less prone to recurrent cardioembolic events and it might be an explanation for why the mortality rate is lower either in cardioembolic or DOAC patient groups.

The median age in the atherothrombotic stroke patient group was 71 (IQR = 64–79) and in the cardioembolic patient group it was 79 (IQR = 72–84). In a study by Beume et al., the median age for patients with large vessel occlusion was 74–75 years [22]; in Stueckelschweiger et al.’s study, it was 74 years [23]. Most of the cardioembolic stroke patient studies report a similar or younger age range of 73–80 [17,18,24].

In our study, the long-term functional outcome was better in cardioembolic stroke patients (as well as in the DOACs group) than in atherothrombotic stroke patients. We also found that there was higher mortality in the atherothrombotic stroke group than in the cardioembolic group. Until now previous studies, including one by Masiliūnas (2022), have concluded that a cardioembolic stroke is associated with poor long-term functional outcomes and high mortality [7]; however, a previous Jurjans et al. study showed similar results comparing 2 ischemic stroke subtypes [8].

In our atherothrombotic stroke group 44 (16.8%) patients received IVT, 3 (1.1%) underwent EVT and 12 (4.6%) received both IVT and EVT. In the cardioembolic stroke group, 69 (17.6%) received IVT, 13 (3.3%) were treated with EVT and 30 (7.7%) with both methods. In the DOAC group, 46 (25.3%) were treated with IVT, 8 (4.4%) with EVT, and 17 (9.3%) with both treatment methods. This can also significantly affect patients’ short-term and long-term outcome.

We discovered that dyslipidemia did not affect the functional outcome of patients. Although hypercholesterolemia and dyslipidemia are known ischemic stroke-modified risk factors, data about the association between dyslipidemia and stroke outcome are controversial. Tian Xu et al. found a strong association between lipid serum levels and ischemic stroke outcomes. In their study, Raja Sheraz Ullah Khan et al. concluded that high cholesterol serum levels worsen ischemic stroke outcomes only in combination with smoking [25,26]; some researchers made the opposite statement, however. Amna Sohail et al. concluded that patients with large vessel occlusion or cardioembolic stroke patients and dyslipidemia had higher mRS scores on admission, but no statistically significant difference in long-term outcome was observed [27]. A similar suggestion was made by Thomas Bowman et al. [28].

Even though SARS-CoV-2 infection is associated with higher thrombotic risk because of many factors (such as hypercoagulation due to the systemic inflammatory process and thrombosis), only 1% of our patients died due to COVID-19 infection. In Adnan Qureshi et al.’s study, only 1.3% of all patients had an ischemic stroke; however, SARS-CoV-2 infection was associated with a higher mortality rate during the intrahospital period and after discharge [29]. Patients with severe comorbidities and prothrombotic states (such as pregnancy or malignancy) have a higher risk for both ischemic stroke and COVID-19 infection complications [30].

The strength of the study includes systematically collected data from all consecutive ischemic stroke patients in a single tertiary center for a 4 year period. We used standard protocols and scales (LV-NIHSS, mRS, CHA2DS2-VASc, HAS-BLED), as well as a standardized follow-up procedure (RFA-A). This is the first long-term study in large vessel occlusion and NVAF stroke survivors with a specific selection of patients receiving DOAC.

This study has some limitations. There is a statistically significant difference in age between the atherothrombotic stroke patients and cardioembolic stroke patients which could impact the results, as patients of an older age are more likely to have an already existing disability or comorbidities.

## 5. Conclusions

Previous studies until now have concluded that cardioembolic stroke is associated with higher mortality and an unfavorable functional outcome. In our study, the cardioembolic stroke group and the DOAC group had a statistically significant higher percentage of patients with congestive heart failure and older age, but their long-term mortality was lower and they achieved independence more often than the atherothrombotic stroke patients. The proper use of anticoagulants shows great improvement in long-term survival rate and functional outcome.

## Figures and Tables

**Figure 1 medicina-58-01323-f001:**
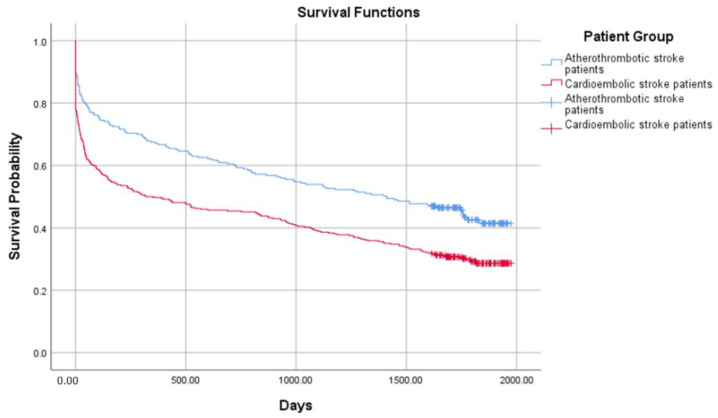
Probability of survival in the atherothrombotic stroke and the cardioembolic stroke groups.

**Figure 2 medicina-58-01323-f002:**
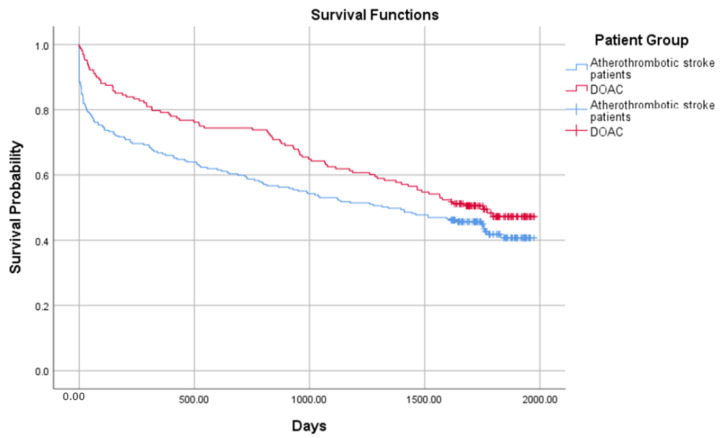
Probability of survival in the atherothrombotic stroke and the direct oral anticoagulant (DOAC) groups.

**Figure 3 medicina-58-01323-f003:**
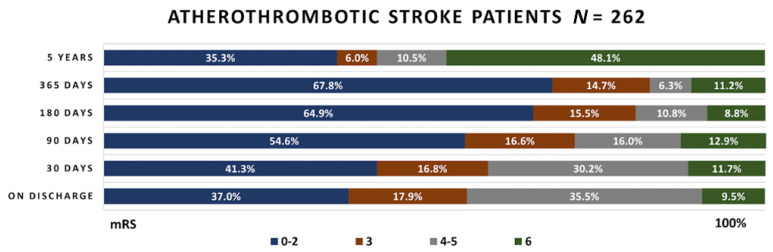
Functional outcome in the atherothrombotic stroke patient group. mRS—modified Rankin scale.

**Figure 4 medicina-58-01323-f004:**
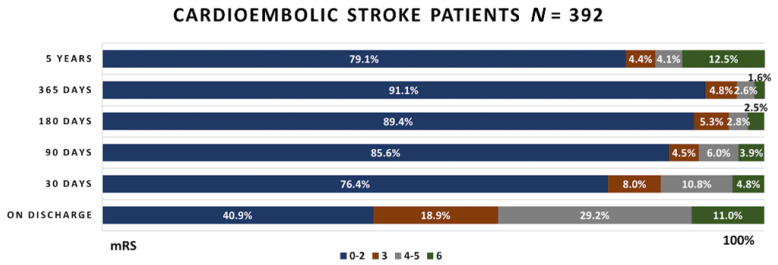
Functional outcome in the cardioembolic stroke patient group.

**Figure 5 medicina-58-01323-f005:**
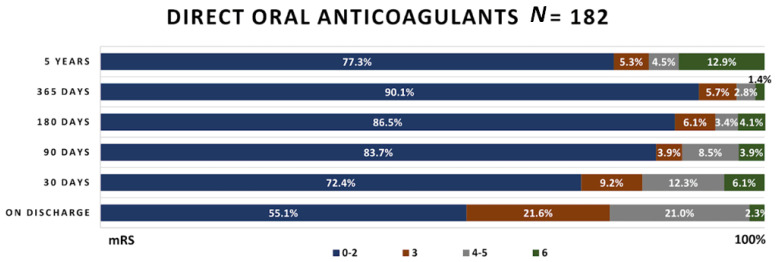
Functional outcome in patients taking DOACs.

**Table 1 medicina-58-01323-t001:** The patients’ treatment data.

Antithrombotic Therapy	Atherothrombotic Stroke Group 40.1% (*N* = 262)	Cardioembolic Stroke Group 59.9% (*N* = 392)	DOAC Group 27.8% (*N* = 182)	*p* (between Atherothrombotic and Cardioembolic Stroke Groups)	*p* (between Atherothrombotic Stroke Group and DOAC Group)
IVT	16.8% (*n* = 44)	17.6% (*n* = 69)	25.3% (*n* = 46)	0.103	0.001
EVT	1.1% (*n* = 3)	3.3% (*n* = 13)	4.4% (*n* = 8)		
IVT + EVT	4.6% (*n* = 12)	7.7% (*n* = 30)	9.3% (*n* = 17)		

IVT—intravenous thrombolysis, EVT—mechanical thrombectomy, IVT + EVT—intravenous thrombolysis and mechanical thrombectomy.

**Table 2 medicina-58-01323-t002:** The patients’ demographic characteristics and neurological deficits.

	Atherothrombotic Stroke Group 40.1% (*N* = 262)	Cardioembolic Stroke Group 59.9% (*N* = 392)	DOAC Group 27.8% (*N* = 182)	*p* (between Atherothrombotic and Cardioembolic Stroke Groups)	*p* (between Atherothrombotic Stroke Group and DOAC Group)
Average age (IQR)	71 (64–79)	79 (72–84)	75 (69–82)	<0.01	<0.001
Coronary heart disease	14.8% (*n* = 35)	16.9% (*n* = 36)	17.0% (*n* = 31)	0.809	0.497
Congestive heart failure	24.8% (*n* = 59)	41.8% (*n* = 89)	42.3% (*n* = 77)	0.001	<0.001
Prior myocardial infarction	8% (*n* = 20)	12% (*n* = 45)	13% (*n* = 13%)	0.107	0.05
Arterial hypertension	84.9% (*n* = 202)	83.1% (*n* = 177)	82.4% (*n* = 150)	0.706	0.616
Dyslipidemia	54.6% (*n* = 143)	26.5% (*n* = 104)	47.8% (*n* = 87)	0.407	0.031
Recurrent stroke or TIA	28.2% (*n* = 67)	25.4% (*n* = 54)	25.3% (*n* = 46)	<0.001	<0.001
Diabetes mellitus	23.1% (*n* = 55)	14.1% (*n* = 30)	15.4% (*n* = 28)	0.084	0.114
Chronic kidney disease	3% (*n* = 7)	6% (*n* = 22)	4% (*n* = 8)	0.73	0.308
Smoking	8% (*n* = 20)	1% (*n* = 4)	1% (*n* = 2)	<0.001	0.002
NIHSS-LV on admission (IQR)	6 (4–10)	8 (4–15)	7 (3–13)	0.379	0.698
NIHSS-LV at discharge (IQR)	3 (2–6)	3 (2–9)	3 (1–5)	0.004	0.009

DOAC—direct oral anticoagulants, IQR—interquartile range, TIA—transitory ischemic attack, NIHSS-LV—The Latvian version of the National Institutes of Health Stroke Scale.

**Table 3 medicina-58-01323-t003:** Cox regression measure among the patient groups.

	Atherothrombotic Stroke vs. Cardioembolic Stroke		Atherothrombotic Stroke vs. Patients Taking DOAC	
Cofactor	Hazard Ratio	*p* Value	Hazard Ratio	*p* Value
Age	1.059	<0.001	1.045	<0.001
Reperfusion therapy	0.844	0.1	0.540	<0.001

## Data Availability

Not applicable.

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
