# Peer review of "Impact of Anticoagulants in Reducing Mortality and Disability in Cardioembolic Stroke Patients"

_medicina, 2022, doi:10.3390/medicina58101323_

Round 1

Reviewer 1 Report

This topic is interesting, but some points need to be addressed. Look at these points:

- Lines 67-68: "The aim of the present study is to compare the functional outcome and mortality in patients with atherothrombotic and cardioembolic stroke, and patients taking DOACs." The aim of this paper should be explain better. What does this paper want to add new to the literature?

- In results, it seems that dyslipidemia does not affect outcome. Discuss this point more.

- Lines 212-213: "A total of 1% of all patients died due to COVID-19 212 infection" Few lines in the discussion section about stroke and Covid-19 must be added. Consider these very interesting paper:  doi: 10.3390/neurolint14020032. PMID: 35645351  ---  doi: 10.33588/rn.7504.2021373

- Lines 225-226: " therefore, the data of the 225 patients do not represent the entire stroke population in our country [14]." So what do authors propose about? Discuss.

- Lines 262-263: "In our study, the five-year functional outcome was better in cardioembolic stroke patients (as well as in the DOACs group) than in atherothrombotic stroke patients" Is there similar paper in pubmed?

- Lines 277-278: "antiplatelet agents are generally seen as a low-cost alternative compared with direct oral anticoagulants" What do authors mean?

- Conclusion is too short. Add some of your results detail as well as any possible further studies authors think about in the next future

- Figures 3 and 4. Improve figures legend.

Reviewer 2 Report

The article evaluates long-term functional outcome and mortality of ischemic stroke in Latvia. The  strength is that the authors are interested in the outcome of their patients, which is likely to be used to analyze where they can improve. I have several comments and questions about the work.

1/ The title states that the data are from Latvia and in the Materials and methods section that they are from Pauls Stradins Clinical University Hospital, Riga, Latvia. If data are only from one hospital, then in the title cannont be that they are from Latvia.

2/ In Materials and methods is written that data are from patients who were admitted to Pauls Stradins Clinical University Hospital in 2017, how the authors can evaluate outcome 5 years after discharge when we have 2022? I don't know exactly when they did the analysis in 2022, but if now is August, then approximately half of the patients from 2017 cannot yet have a 5-year outcome.

3/ The authors devided patients with cardioembolic stroke to 2 groups, patients treated with DOACs and without DOACs. It could be better to use patients with ischemic stroke and AF treated with DOACs.

4/ There is no any information if patients were tretaed with intravenous thrombolysis or mechanical thrombectomy, which can significantly affect the patient's outcome.

5/ In the group with atherothrombotic stroke did some patients also use dual antiplatelet therapy, which is recommended for minor stroke in the first 21 days after a stroke?

6/ In the group with atherothrombotic stroke were there also patients with hemodynamically severe ICA stenosis, if so, did they have carotid endartherectomy done?

7/ On page 6 there are two figures 3, probably cardioembolic stroke patients should by Fig. 4, and then Figure about patients with DOAC should be 5. Subsequently, it must be corrected in the text - page 5.

8/ I would recommend the text after Fig. 4 on page 5, which continues on page 6, should be completely omitted, because all the results are shown in the figures.

Round 2

Reviewer 1 Report

Authors solved all my criticisms.

Reviewer 2 Report

The authors accepted most of the comments and refined the data.

I have few comments:

1/ The authors changed 5 years to 4+, but 4+ can be 5 years, 6 years, etc. It should be clearly stated how long the patients were followed.

2/ The question in point 3 was different than answer. Although the question was not whether they used DOACs or vitamin-K antagonists, the authors stated that they used vitamin-K antagonists, which was not in the original text. But in the title they write:

"Impact of direct oral anticoagulants in reducing mortality and disability in cardioembolic stroke patients" -

so or they should exclude patients with vitamin-K antagonists, or change the title:

„Impact of anticoagulants in reducing mortality and disability in cardioembolic stroke patients“

Point 3: The authors devided patients with cardioembolic stroke to 2 groups, patients treated with DOACs and without DOACs. It could be better to use patients with ischemic stroke and AF treated with DOACs.

Response 3: Some of our patients are receiving vitamin K antagonists. Would you like us to remove them from the cardioembolic patient group from our research? Would it not be a selective patient inclusion?

4/ The authors added information about IVT and ET in both groups, but the statistical analysis if only use od OACs, or also acute treatment influenced the outcome should be done.

5/ In the raw 155 is probably typo - IAC endarterectomy.
